# Current Achievements and Future Challenges of Genotype-Dependent Somatic Embryogenesis Techniques in *Hevea brasiliensis*

**Xiaoyi Wang** [1,2,†], **Xiaochuan Gu** [1,2,†], **Zhengwei Xu** [1,2], **Zhaochen Yin** [3], **Xianfeng Yang** [1,2], **Rong Lin** [3], **Quannan Zhou** [1,2], **Huasun Huang** [1,2,*] **and Tiandai Huang** [1,2,*]

1  Key Laboratory of Biology and Genetic Resources of Rubber Tree, Ministry of Agriculture and Rural Affairs, National Key Laboratory for Tropical Crop Breeding, Rubber Research Institute, Chinese Academy of Tropical Agricultural Sciences, Haikou 571101, China; micky_xiao@catas.cn (X.W.); xjsgxc@catas.cn (X.G.); xzw861127@outlook.com (Z.X.); shouyiweida@catas.cn (X.Y.); zhouquannan2004@163.com (Q.Z.)

2  Haikou Key Laboratory of Tropical Plant Seedling Innovation, Haikou 571101, China

3  College of Tropical Crops, Yunnan Agricultural University, Pu'er 665000, China; yzc2429698957@outlook.com (Z.Y.); 2006046@ynau.edu.cn (R.L.)

*  Correspondence: huanghuasun@catas.cn (H.H.); huangtiandai@catas.cn (T.H.)

†  These authors contributed equally to this work.

**Abstract:** The rubber tree (*Hevea brasiliensis*) is the most important commercial plant for producing natural rubber. Immature seed inner integument and anther-derived somatic embryogenesis techniques play a crucial role in the in vitro large-scale propagation and genetic transformation of the rubber tree. However, somatic embryogenesis is highly genotype-dependent, that is, only a limited number of *H. brasiliensis* genotypes, such as CATAS73397, CATAS917, and PB260, can be efficiently induced by somatic embryogenesis and used for large-scale propagation or transformation. The genotype dependence of the somatic embryogenesis technique is a conundrum for the application of *Hevea* biotechnology in most commercially important cultivars, such as Reken628 and CATAS879. Previous studies have shown that several somatic embryogenesis regulators can overcome genotype dependence and enhance the transformation and regeneration efficiency of recalcitrant plants and cultivars. In this review, we first describe the relevant successful applications of somatic embryogenesis technology in seedling production and genetic modification of *H. brasiliensis*. Second, we discuss the genotype dependence of somatic embryogenesis as the major challenge currently. Third, we summarize the recent significant advances in the understanding of the molecular mechanisms underlying somatic embryogenesis in other plants. Finally, we suggest a roadmap for using somatic embryogenesis regulatory genes to facilitate genotype-dependent somatic embryogenesis technology in *H. brasiliensis*.

**Keywords:** vegetative propagation; genetic modification; genotype dependence; somatic embryogenesis




## 1. Introduction

The rubber tree (*Hevea brasiliensis*) is a tall perennial tree belonging to the Euphorbiaceae family. It is a typical tropical rainforest species, native to the Amazon basin, and is currently found in over 40 countries and regions in Asia, Africa, Oceania, and Latin America [1]. The rubber tree is the only source of commercially produced high-quality natural latex (*cis*-1, 4-polyisoprene) among over 2500 rubber-producing plants such as rubber dandelion (*Taraxacum kok-saghyz*) and Eucommiae cortex (*Eucommia ulmoides*) [2,3]. Rubber dandelions can produce natural rubber with a higher molecular weight of up to 20% of their dry weight, and they can be widely planted in temperate zones [4]. Eucommiae cortex accumulates a large amount of elastic trans-1,4-polyisoprene in its bark, pericarp, and leaf [5]. However, these rubber-producing plants cannot produce substantial amounts of high-quality rubber [3]. Natural latex has superior physical and chemical properties,

which include excellent elasticity, resilience, and abrasion resistance, making natural latex unable to be replaced by synthetic rubber in many industrial utilizations, such as in national defense, medicine, aerospace, navigation, and automobile manufacturing industries [3].

In recent years, the demand for natural latex has increased with the rapid development of global industry and the military, as well as changes in the international situation and resource constraints. According to the Association of Natural Rubber Producing Countries (ANRPC), global natural rubber production in 2021 was 13.8 million tons, and the demand was 14.1 million tons, with a yield gap of 0.3 million tons [6]. More than a century of traditional breeding has resulted in a significant increase in rubber production, from 650 kg ha$^{-1}$ obtained from unselected seedlings of *H. brasiliensis* during the 1920s to 2500 kg ha$^{-1}$ obtained from superior cultivars during the 1990s [7]. However, the current yield is still far below the hypothetical yield of 7000–12,000 kg ha$^{-1}$ for the rubber tree [8]. Due to constraints, such as a long breeding cycle and low breeding efficiency, the genetic improvement of high-yield cultivars through traditional hybrid breeding faces challenges in *H. brasiliensis*. Thus, there is an urgent need to introduce new breeding technologies to accelerate germplasm innovation for latex yield and to obtain high-quality rubber tree seedlings.

With the rapid development of biotechnology, genetic modification may be a highly efficient method to develop a high-yield and stress-resistant rubber tree in a short period of time. The successful application of genetic modification relies on the ability to regenerate transgenic plants. To achieve a high multiplication rate, somatic embryogenesis (SE) technology has been proven to be a promising regeneration system for genetic modification of *H. brasiliensis* [9–11]. Meanwhile, the plantlets derived from the somatic embryos of mature plants through somatic embryogenesis are suitable for rejuvenating selected mature clones with their own roots, and these self-rooting juvenile clones (SRJCs) demonstrated higher yield, faster growth, and higher stress resistance than mature plants [12–15]. Therefore, SE technology is also a promising vegetative propagation system in *H. brasiliensis*. However, due to genotype dependence, SE technologies have only been established for a few varieties of *H. brasiliensis*, which restricts the application of vegetative propagation systems and genetic modification technology [14].

This review summarizes research progress in our understanding of SE in *H. brasiliensis*, including: (I) a brief overview of current SE technologies used in seedling production; (II) genetic modification technical breakthroughs based on SE technologies, which can carry out precise characteristic improvement; and (III) future challenges and suggestions of genotype dependence in the application of SE technologies in *H. brasiliensis*. Finally, future perspectives on the establishment of genotype-independent SE technologies that will facilitate the application of CRISPR/Cas-based genome editing or other biotechnological methods in *H. brasiliensis* are discussed.

## 2. Somatic Embryogenesis in *H. brasiliensis*

In vitro, plant somatic cells are capable of regenerating complete plants under appropriate conditions, a phenomenon known as cell totipotency [16]. SE is a notable representation of cell totipotency, which involves the developmental reprogramming and dedifferentiation of somatic cells to regain cell totipotency under specific culture conditions [17]. The development of somatic embryos from one or more totipotent embryogenic stem cells includes the following stages: globular embryoid, heart-shaped embryoid, torpedo-shaped embryoid, cotyledonary embryoid, and matured cotyledonary embryoid, which resembles that of zygotic embryos [18].

To date, the capacity to regenerate an entire plant through SE has been reported in various plants, including *Arabidopsis*, *Gossypium*, and *Liquidambar* hybrids, which has not only provided an effective method for virus-free plant propagation but has also greatly assisted gene transformation in many plant species [19,20]. Plant regeneration systems via SE in *H. brasiliensis* have been established using different kinds of explants, including anther-

derived callus [12], inner integument-derived callus [13], secondary somatic embryos [14], leaf [21], embryogenic cell suspension lines [22], and protoplasts [23].

SE can occur in two different ways: direct SE, which is a process of directly inducing somatic embryos on the surface of explants without involving an intermediate callus, or indirect SE from callus [16]. In *H. brasiliensis*, the regeneration systems from the anther and immature seed internal integument were established mainly through the indirect SE process, which includes two main stages: the embryogenic callus induction stage for the acquisition of cell totipotency under exogenous auxin and somatic embryo formation and the development stage under culture without auxin [24]. The regeneration system of cotyledonary somatic embryos was established mainly through the direct SE process without a callus phase [14].

## 3. Application of Somatic Embryogenesis Technologies in *H. brasiliensis* Seedling Production

The effective natural rubber production of *H. brasiliensis* is determined by seedling quality, which will affect the growth, rubber yield, and stress resistance for over 30 years after planting [25]. The development of seedling propagation methods has gone through three stages in *H. brasiliensis*. The first stage used the seeds of *H. brasiliensis* as the source of the seedlings. *Hevea brasiliensis* is a monoecious and cross-pollinated woody plant with a highly heterozygous genome; thus, the characteristics of seedlings developed from seeds are very different. The second stage, the budding technique, was developed in the 1920s for clonal reproduction of *H. brasiliensis* with desirable properties [26]. Although grafted seedlings have improved the characteristic consistency of rubber trees, grafting success primarily depends on graft compatibility between the rootstock and scion. Subsequently, researchers have found that the origin of axillary buds is very important. They found that the axillary buds originate from the stems of young seedlings or from the base of the tree, and the grafted seedlings will develop through a juvenile stage and present better performance in rubber yield, trunk girth, and laticifer number than those from the crowns of mature trees [27]. Following this line of thought, in the 1980s, researchers from multiple countries, including Malaysia, France, and China, obtained juvenile plantings developed from somatic embryos of mature plants, giving rise to a new era in the third generation of planting material [11–13].

In *Hevea*, only immature seed integuments and anthers can be used to induce embryogenic callus efficiently [11–13,28]. Initially, immature seed integument-derived explants were used to produce embryogenic callus that could be used for primary SE induction [29]. Subsequently, integument-derived explants were used to produce embryogenic friable callus, which could be maintained by repeated successive subcultures for long-term SE [11,29,30]. From 1992 to 2002, 65 ha of field trials in three countries with a statistical design were conducted to assess the genetic fidelity, rubber yield, growth, and development of integument-derived plants, in comparison with mature budded plants [31]. Although SRJCs or budded rejuvenated clones from primary SE exhibited more excellent traits in growth and yield than mature budded plants and a low risk of somaclonal variation, primary SE is not suitable for mass propagation. In contrast, more than 17,000 SRJCs or budded rejuvenated clones from long-term SE of rubber tree clone PB260 showed unsatisfactory growth compared to mature budded plants and had a high risk of somaclonal variation, which restricted its potential in mass propagation [31]. To avoid a high risk of somaclonal variation and loss of rejuvenated competence during long-term SE, Lardet et al. developed a reliable cryoconservation technique for friable embryogenic callus lines [32]. The technique showed that pre-culture on a medium with 0 or 1 mM of $CaCl_2$ before cryopreservation promoted post-thaw callus growth recovery. This is because a low concentration of exogenous calcium leads to a decrease in the endogenous calcium content of the callus and subsequently induces cryotolerance and post-thaw embryogenic competence in the callus [32].

Compared with internal integuments, using anthers as explants has several advantages, including a large number of flowers, and anther-derived somatic embryo seedlings have demonstrated a high rubber yield, quick growth, and high stress resistance compared to grafted seedlings of the same variety in the field [33]. Subsequently, based on anther-derived cotyledonary somatic embryos, the micropropagation technique of SRJCs through secondary SE in *H. brasiliensis* was established in 2010, which can supply large-scale production of high-quality planting material in a cyclic routine [14]. This micropropagation technology is a rapid and efficient method for mass propagation of *H. brasiliensis*, which can be done all year round [14]. The genetic fidelity of the secondary SE regenerated plants from 10 multiplication cycles compared with the mother tree was reported. The study found that the regenerated plants had no chromosome number variation and a low risk of somaclonal variation, which indicates high genetic stability during secondary SE and can be used for commercial mass production [20]. Long-term and short-term maintenance of anther-derived cultures has been reported by several researchers. Zhou et al. successfully set up the long-term cryopreservation technique of anther-derived callus using the vitrification method, and post-thaw callus showed 71.7% viability [34]. Zhou et al. found that exogenous antioxidant pretreatment of anther-derived embryogenic callus before low-temperature storage (4–8 °C) protected the cell activities of embryogenic calli and enhanced their plant regeneration capacity [35].

The micropropagation of SRJCs by secondary embryogenesis was established by following three steps: first, the induction of primary embryos was carried out; second, primary embryos were cut into fragments for secondary embryogenesis, and this step can be repeated over and over again in cyclic form; and third, plants were regenerated from secondary embryos (Figure 1) [14]. At present, the first large-scale production line of SRJCs of *H. brasiliensis* has been established in China. Through cyclic secondary embryogenesis, the annual production of CATAS7-33-97 SRJCs is over 1 million plants [36]. The performance of SRJCs has been better than that of grafted seedlings of the same variety in regional trials on more than 1333 ha, showing better uniformity of forest form, 20% faster growth, 20% higher yield, superior cold resistance, and wind resistance [12,37,38]. Therefore, the micropropagation of the SRJC system by secondary embryogenesis can not only produce high-quality seedlings of *H. brasiliensis* but can also increase natural rubber yield worldwide.

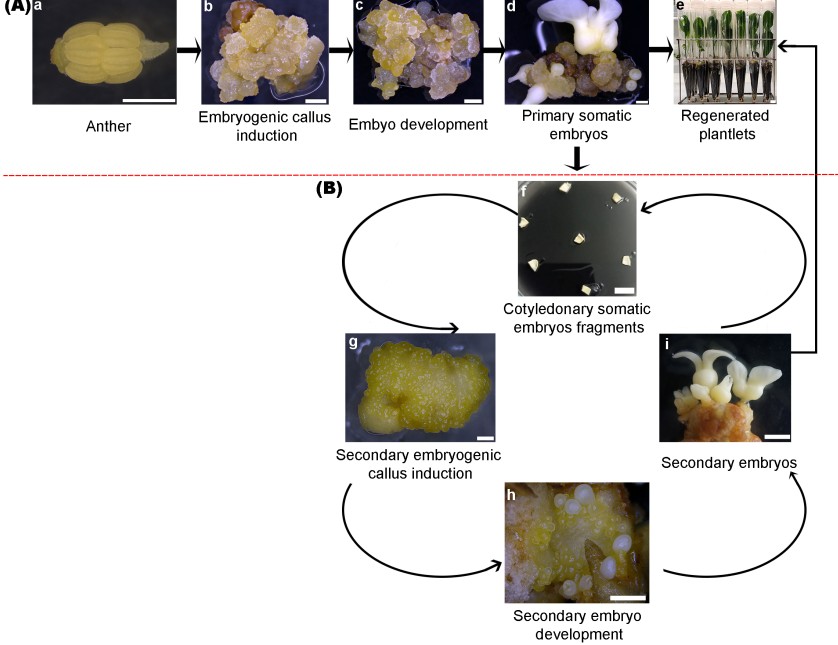

**Figure 1.** Anther-derived somatic embryogenesis in *Hevea brasiliensis* for producing self-rooting juvenile clones. (**A**) Primary somatic embryogenesis. First, anthers (a) were cultured on a callogenesis

medium for 50 days to induce embryogenic calli (b). Second, the embryogenic calli were transferred onto an embryogenesis medium for 30 days to induce primary somatic embryos (c and d). Finally, the primary embryos were transferred onto a regeneration medium to produce regenerated plantlets (e). (**B**) Secondary embryogenesis. First, the primary somatic embryos were cut into fragments (f) and cultured on a callogenesis medium for 30 days to induce direct somatic embryogenesis (g). Second, the secondary embryogenic calli (g) were transferred onto an embryogenesis medium for 60 days to induce secondary somatic embryos (h and i). Finally, the secondary embryos were transferred onto a regeneration medium to produce regenerated plantlets (e). The secondary embryogenesis process can be repeated over and over again in cyclic form.

When plantlets are regenerated from somatic embryos of anthers and integuments of *H. brasiliensis*, it is practicable to rejuvenate selected mature plants with their own roots. Therefore, somatic embryo-derived plants grow stronger. To understand the molecular mechanism of better performance of SRJCs, researchers have carried out a comparative transcriptome analysis for SRJCs and donor mature clones. They found that the genes related to latex metabolism and flow were upregulated in SRJCs compared to donor mature clones, which led to increased yield in SRJCs [39]. Genes related to epigenetic modification enzymes are differentially expressed between SRJCs and donor clones, which means that epigenetic modification may play an important role in the higher yield and resistance of embryo-derived plants [39].

## 4. Application of Somatic Embryogenesis Technologies in the Genetic Modification of *H. brasiliensis*

The improvement and innovation of *H. brasiliensis* varieties are fundamental to the sustainable development of the natural rubber industry globally. Until the 1990s, many high-yielding and high-quality varieties had been developed via natural selection or artificial crossbreeding in *H. brasiliensis*, which resulted in a significant improvement in latex yield [8]. However, rubber production is still far below the suggested theoretical yield [40]. There are two main reasons why it is difficult to further improve yield through traditional breeding approaches in *H. brasiliensis*. First, the domesticated cultivars originated from a small number of wild individuals, which led to low genetic diversity for cultivated *H. brasiliensis* germplasm [41]. Second, due to the genetic burden of removing unfavorable alleles that are tightly linked to favorable alleles, the highly heterozygous genome, and the long breeding cycle, it is difficult to introduce excellent characteristics from wild germplasm into commercial varieties [42]. The application of genetic manipulation technologies offers promise to improve breeding efficiency for *H. brasiliensis* by aggregating excellent genes.

Genetic transformation is a crucial step in applying genetic manipulation technologies to basic and applied plant science research. The success of genetic transformation primarily relies on the efficiency of gene delivery into plant cells and the ability of plant regeneration. *Agrobacterium*-mediated transformation is the preferred method for delivering a foreign gene into *H. brasiliensis* cells. Previous studies have made steady progress in developing a highly efficient genetic transformation system in all facets, such as the target explants, geno-types, transformation methods, tissue culture protocols, transformation vectors, hormone concentrations, and selection strategies (Table 1).

During *H. brasiliensis* genetic transformation, regeneration is generally performed via SE based on friable embryogenic callus and anther callus or secondary embryogenesis based on mature cotyledonary somatic embryos [9,10,43]. Since friable embryogenic callus has been developed from the inner integument of immature fruit [11,30,44], efficient *Agrobacterium tumefaciens*-mediated genetic transformation of embryogenic calli in *H. brasiliensis* has been established [43]. To date, *HbCuZnSOD*, *HbERF-IX c5*, *HMGR1*, and *EcGSH1* transgenic plants have been obtained using friable embryogenic callus as explants, which show enhanced tolerance to abiotic stress [45–47]. Anther-derived callus and somatic embryos are also excellent explants for *H. brasiliensis* genetic transformation [14]. Various genes have been successfully transferred into *H. brasiliensis* through anther-derived callus or somatic embryos, such as *GUS*, *HbSOD*, *HSA*, *GAI*, Bean *Chitinase*, tobacco *β-1,3-Glucase*,

and *HbAN2* (Table 1) [9,10,48–53]. The most efficient transformation system using cotyledonary somatic embryos of CATAS7-33-97 as explants was developed by Udayabhanu et al. (Figure 2), which optimized several crucial factors, such as sonication and co-cultivation conditions, antibiotic concentration, and *Agrobacterium* concentration [54]. However, due to the genotype dependency of SE, the applicability of these transformation methods is limited [55].

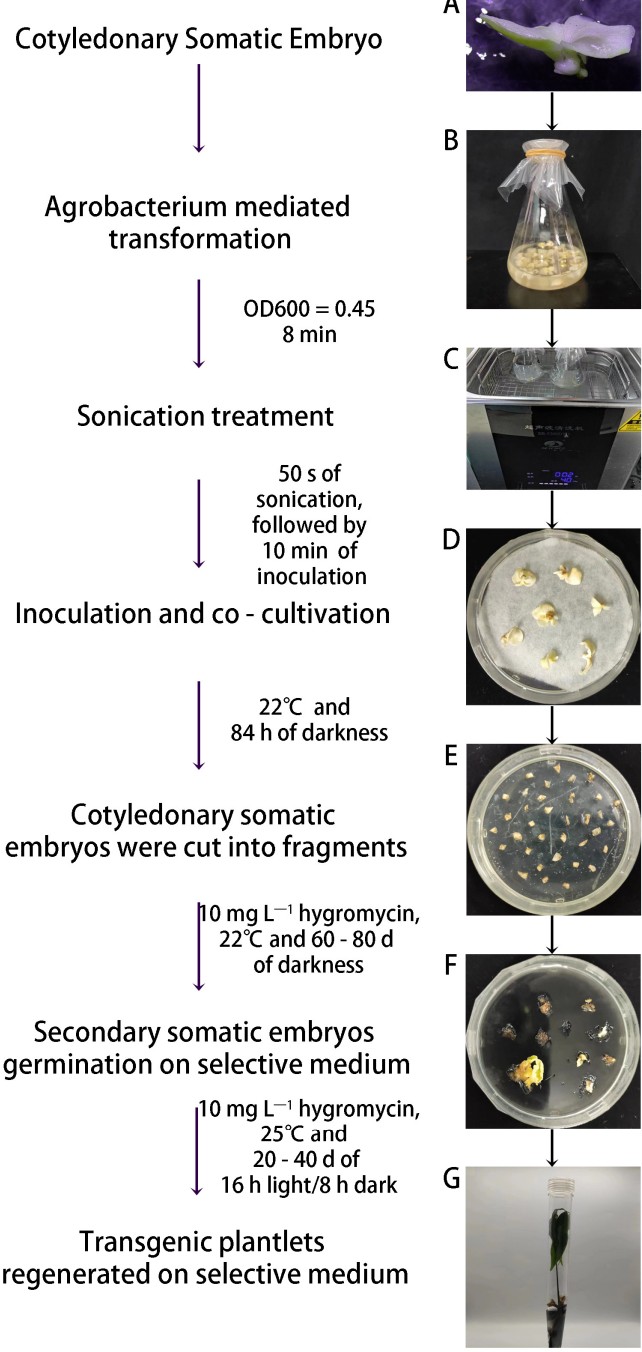

**Figure 2.** Schematic representation of a highly effective genetic transformation method for *Hevea brasiliensis* using somatic embryos. (**A**) Primary and secondary cotyledonary somatic embryos were used as explants; (**B**,**C**) Mixture of somatic embryos and *Agrobacterium* cells treated using sonication; (**D**) Co-cultivation for three days on medium with acetosyringone under dark conditions at 22 °C; (**E**) Cotyledonary somatic embryos were cut into fragments and transferred onto selective medium to induce secondary embryogenic callus; (**F**,**G**) Transgenic plants were developed by subculture and selection.

**Table 1.** Summary of genes transformed in *Hevea brasiliensis* transgenic research.

| Genotype | Explant Tissue | *Agrobacterium* Strain | Promoter | Plasmid | Transformed Gene | Transformation Efficiency | References |
|---|---|---|---|---|---|---|---|
| GL-1 | Anther-derived callus | Particle gun method | CaMV 35s | pBI221; pMON9793; pDE10; pHP23. | *gus*; *npt II*; *cat* | 80 callus, 2 transgenic plantlets | [50] |
| GL-1 | Anther-derived callus | GV2260 | CaMV 35s | p35SGUSINT | *gus*; *npt II* | 35 transgenic callus obtained from 438 callus (8.0% transformation efficiency); 2 regenerated transgenic plantlets obtained from 65 resistant embryoids (3.1% regeneration efficiency) | [56] |
| GL-1 | Anther-derived callus | GV2260 | CaMV 35s | pLGMR: pToK47 | *HSA* | 423 transgenic callus obtained from 5711 callus (7.4% transformation efficiency); 102 regenerated transgenic plantlets obtained from 1446 resistant embryoids (7.1% regeneration efficiency) | [51] |
| RRII 105 | Anther-derived callus | EHA101 | CaMV 35s | pDU96.2144 | *HbSOD* | 4% transformation efficiency; 7.1% regeneration efficiency | [48] |
| PB260 | Inner integument tissues of immature fruit-derived friable callus | EHA105 | CaMV 35s | pCAMBIA2301 | *gus*; *npt II* | More than 1695–5264 transgenic events per gram of callus; 372 transgenic plantlets obtained | [43,57] |
| RRII 105 | Leaf-derived embryogenic callus | LBA4404 | CaMV 35s | - | *TB antigen* | 60% transformation efficiency | [58] |
| PB260 | Inner integument tissues of immature fruit-derived friable callus | EHA105 | pHEV2.1 | pCAMBIA1381Z | *gus*; | 14 independent transgenic callus lines | [59] |
| PB260 | Inner integument tissues of immature fruit-derived friable callus | EHA105 | CaMV 35s | pCAMBIA2300 | *GFP*; *EcGSH1* | 23 independent transgenic callus lines | [60] |
| PB260 | Inner integument tissues of immature fruit-derived friable callus | EHA105 | CaMV 35s | pCAMBIA2300 | *HbCuZnSOD-GFP* | 72 independent transgenic callus lines; 8.4–9.5 regenerated transgenic plantlets per gram of callus | [47] |
| CATAS 7-33-97 | Anther-derived callus | EHA105; | CaMV 35s | pCAMBIA2301 | *gus*; | 11 transgenic plantlets from 2.2 million anther callus | [9] |
| CATAS 7-33-97 | Anther-derived callus | EHA105; LBA4404 | CaMV 35s | pBLGC | *Bean Chitinase*; *Tobaccoβ-1,3-Glucase gene* | 0.33% transformation efficiency | [52] |
| CATAS 87-6-62 | Cotyledonary somatic embryos | EHA105 | CaMV 35s | pCAMBIA2301 | *gus* | 0.5% transformation efficiency | [10] |

**Table 1.** *Cont.*

| Genotype | Explant Tissue | *Agrobacterium* Strain | Promoter | Plasmid | Transformed Gene | Transformation Efficiency | References |
|---|---|---|---|---|---|---|---|
| RRII 105 | Inner integument tissues of immature fruit-derived friable callus | - | Super promoter | pBIB | *HMGR1* | 44% regeneration efficiency | [61] |
| PB260 | Inner integument tissues of immature fruit-derived friable callus | EHA105 | CaMV 35s; HEV2.1 | pCamway | *HbERF-IX c5* | 16 transgenic callus obtained from 300 callus (5.3% transformation efficiency); 2–223 resistant embryos per gram of callus; 1–6 regenerated transgenic plantlets per gram of callus | [45] |
| PB260 | Inner integument tissues of immature fruit-derived friable callus | EHA105 | CaMV 35s; HEV2.1 | pCAMBIA2300-GFP-EcGSH1 | *EcGSH1* | 13 transgenic callus lines; 77–1039 resistant embryos per gram of callus; 0–35 regenerated transgenic plantlets per gram of callus | [46] |
| Haiken 2 | Anther-derived callus | Particle bombardment | CaMV 35s | pBI121 | *GAI-GUS* | 149 resistant embryoids obtained from 8771 callus (1.7% transformation efficiency); 8 regenerated transgenic plantlets obtained from 149 resistant embryoids (5.4% regeneration efficiency) | [49] |
| CATAS 7-33-97 | Cotyledonary somatic embryos | EHA105; | CaMV 35s | pCAMBIA2300 | *HbAN2*; *gus* | - | [53] |
| CATAS 7-33-97 | Cotyledonary somatic embryos | EHA105 | CaMV 35s | pCAMBIA2301 | *gus* | - | [54] |

Cat, chloramphenicol acetyl transferase; CuZnSOD, copper–zinc superoxide dismutase; EcGSH1, gamma-glutamylcysteine synthetase of Escherichia coli; GFP, green fluorescent protein; gus, β-glucuronidase; HAS, human serum albumin; HbAN2, Hevea brasiliensis anthocyanin 2; HbSOD, superoxide dismutase; HMGR1, 3-methylglutaryl Co-enzyme A reductase 1; Npt, neomycin phosphotransferase; pHEV2.1, the promoter of the HEVEIN2.1 gene.

With the development of genetic transformation methods and the publication of the *H. brasiliensis* genome, the clustered regularly interspaced short palindromic repeats/CRISPR-associated protein 9 (CRISPR/Cas9) system as a precise gene-editing tool will be helpful for functional genomic research and breeding. The CRISPR/Cas9 system includes two important components: an RNA-guided Cas9 endonuclease that precisely cleaves the target genomic sites and short-guide RNAs (sgRNAs) that specifically direct the Cas9 to the target genomic site [62]. The most significant and attractive advantages of the CRISPR/Cas9 system are threefold: specific and predictable genome editing, multiple genes can be simultaneously edited, and Cas-free transgenic plants can be obtained [63]. In 2021, China successfully established the CRISPR/Cas9 genome editing system in *H. brasiliensis* by employing the *HbU6* promoter and the *CaMV35S* promoter for driving sgRNA and Cas9 transcription [64]. To obtain Cas-free transgenic plants, a Cas9-free genome editing system was developed in *H. brasiliensis* by directly delivering preassembled Cas9 protein-gRNA ribonucleoproteins (RNPs) into rubber tree protoplasts [65]. After genome editing, the RNP complexes were rapidly degraded by endogenous cellular proteases. Therefore, the application of various CRISPR/Cas 9 systems and the setting up of protoplast regeneration systems in *H. brasiliensis* may not only lead to germplasm innovation but may also address regulatory issues and public concerns for genetically modified plants.

## 5. Genotype Dependence Is a Major Problem in the Application of Somatic Embryogenesis Technologies in *H. brasiliensis*

The acquisition of SE ability depends on the pluripotency of plant cells, which is affected by multiple factors, including genotype, hormone concentration, and culture conditions. The efficiency of integument-derived or anther-derived primary SE and secondary SE in *H. brasiliensis* is summarized in Table 2. Genotype is the most decisive factor for SE in *Hevea* in both integument and anther.

**Table 2.** Efficiency of primary somatic embryogenesis and secondary somatic embryogenesis in *Hevea brasiliensis*.

| | Primary Somatic Embryogenesis | | | |
| --- | --- | --- | --- | --- |
| Explant | Callus Induction Rate | Embryogenic Callus Induction Rate | Somatic Embryo Induction Rate | Clones with High Somatic Embryogenesis Ability |
| Immature seed inner integument | 87%–99% | 36%–51% | 43%–209% (number of somatic embryos/total number of explants × 100%) | PB217, PB260, PB280, PB310, PR107, RRIM600 |
| Anther | 53.7%–96.5% | 26.8%–95% | 0%–23.1% (number of callus-forming cotyledonary somatic embryos/total number of callus inoculated × 100%) | PB86, RRIM600, CATAS88-13, Haiken1, Dafeng95, Haiken2, CATAS917, CATAS73397, CATAS879, CATAS918, Xuyu3, Xuyu141-2 |
| | Secondary somatic embryogenesis | | | |
| Explant | Annual multiplication coefficient | | | Clones |
| Cotyledonary somatic embryo | 696–2525 | | | CATAS73397, CATAS88-13 |

The SE ability was analyzed for the immature seed inner integument of 18 main rubber tree cultivars [29,66]. PB217, PB260, PB280, PB310, PR107, and RRIM600 have high somatic embryo induction rates. The callus induction rate by the primary SE method ranges between 87% and 99%. The embryogenic callus induction rate ranges from 36%

to 51% (the embryogenic callus induction rate = number of embryogenic callus/total number of explants × 100%; the callus bearing one or more somatic embryos, irrespective of the development stage, is defined as an embryogenic callus). The percentage of total embryos ranges from 43% to 209% (the percentage of total embryos = number of somatic embryos/total number of explants × 100%) [29,66].

To analyze the effect of different *H. brasiliensis* varieties on anther-derived SE, anthers were collected from different varieties. The results showed that the SE ability is closely related to parental origin. When both parents (such as GT1, Tianren31-45, 93-114, and RRIM513) have a low somatic embryo induction rate, most of their hybrid offspring (such as Yunyan77-2, Yunyan77-4, Yunyan73-46, Baoting1-285, Zhanshi32713, and CATAS217) have a low rate. However, when one of the parents (such as PB86, RRIM600, CATAS88-13, and Haiken1) has a high SE ability, a filial $F_1$ hybrid (such as Dafeng95, Haiken2, CATAS917, CATAS73397, CATAS879, CATAS918, Xuyu3, and Xuyu141-2) often outperforms the inbred parents in SE ability [55]. Following the results of clones with high SE ability, the callus induction rate ranges between 53.7% and 96.5%. The embryogenic callus induction rate ranges from 26.8% to 95% (the embryogenic callus induction rate = number of embryogenic callus/total number of callus inoculated × 100%). The cotyledon somatic embryo induction rate ranges from 0% to 23.1% (the cotyledon somatic embryo induction rate = number of callus-forming cotyledonary somatic embryos/total number of callus inoculated × 100%) [55].

Micropropagation of SRJCs through secondary SE has been established in CATAS7-33-97 and CATAS88-13. The embryo multiplication coefficients of CATAS7-33-97 and CATAS88-13 by the secondary SE method are 13.2 and 6.9, respectively, in the first cycle. Subsequently, the coefficients of CATAS7-33-97 and CATAS88-13 increase to 15 and 11.6, respectively, in the second cycle and remain stable in the third cycle. The plant regeneration ratios of CATAS7-33-97 and CATAS88-13 by the secondary SE method are 85% and 75%, respectively. Three cycles of multiplication can be performed per year, which means that the annual multiplication coefficients of CATAS7-33-97 and CATAS88-13 are 2524.5 and 696.3, respectively [14]. However, secondary SE also has the disadvantage of genotype dependence.

The obvious genotypic differences in the SE ability have become a troublesome issue that restricts the application of the third-generation somatic embryo seedling planting technology, genetic transformation, and gene editing technology in *H. brasiliensis*. Previous studies have shown that the SE ability can be improved by optimizing the culture conditions, hormone concentration, abiotic stress, and overexpression of developmental regulators [11,16,55,67]. Overexpression of developmental regulators in *Arabidopsis*, maize, rice, sorghum, and citrus can effectively improve plant SE, genetic transformation, and gene editing efficiency [68]. For example, *Leafy cotyledon 1* (*LEC1*), *LEC2*, and *BABY BOOM* (*BBM*) are specifically expressed during zygotic embryogenesis and development [69–72]; *WUSCHEL* (*WUS*) is specifically expressed in pluripotent stem cells [73]; and *Growth-Regulating Factor (GRF)* and its cofactor *GRF-interacting factor (GIF)* regulate plant growth and development [68]. Therefore, due to species specificity, it is necessary to understand the molecular mechanism and identify the key genes associated with SE in *H. brasiliensis*, which can be used for targeted breeding to improve SE.

## 6. The Somatic Embryogenesis Mechanism in Plants

After the first in vitro regeneration of carrot was performed through SE, the physiological, cellular, and molecular mechanisms of SE have been studied in plants [16,74]. These studies found that optimization of hormone concentration and overexpression of SE-related transcription factors could effectively improve the SE ability.

Hormones play an important role in SE induction. In general, somatic embryo formation requires exogenous auxin addition and auxin removal in two successive steps. First, cell dedifferentiation, callus formation, and embryonic competence acquisition are induced by culture on auxin-rich callus-inducing medium, and then, the callus is transferred to

an auxin-free medium to produce somatic embryos [75]. After the removal of auxin from the medium, auxin biosynthesis, auxin gradient establishment, and PIN1-mediated polar auxin transport in the embryonic callus are activated and are essential for SE and the induction of *WUS* expression in the shoot apical meristem (SAM) [73]. In contrast, the establishment of the cytokinin response in the callus is correlated with SE and induces *WOX5* expression at the embryonic root apical meristem (RAM) [76]. Therefore, auxin and cytokinin signaling play distinct functions in SAM and RAM induction and shoot–root axis establishment during early SE [75]. After the removal of exogenous auxin from the medium, the genes involved in ethylene biosynthesis and the signaling pathway are downregulated, and the ethylene level decreases. The decreased level of ethylene, in turn, promotes the expression of *YUCCAs* that might be involved in local auxin biosynthesis and subsequent auxin distribution, which are critical for somatic embryo induction [77]. In addition, abscisic acid (ABA) negatively regulates SE by inhibiting polar auxin transport and distribution in embryogenic callus [78]. Even though adjusting the hormone levels in the culture medium has been widely used for optimizing SE technologies in some plants, an efficient and genotype-independent SE technology has not been established in most plants.

Over the past few decades, the molecular mechanism of SE in plants has been intensive, and some developmental regulators involved in SE are already known, such as *WUS*, *BBM*, *LEC1/2*, *GRF*, and *GIFs* [79,80]. The ectopic expression of these regulators is associated with improving the developmental reprogramming of somatic cells to a pluripotent state and the reacquisition of an embryonic or meristematic development fate, and the loss-of-function mutations in these regulators significantly impair regeneration [75]. Recently, several developmental regulators have been used to improve the SE or organogenesis ability in monocots and dicots [68,69,71,81–83]. For example, overexpression of the maize *BBM* and *WUS2* genes can significantly improve SE and transformation frequencies in several recalcitrant maize inbred lines, sorghum, sugarcane, and rice [81,84]. Debernardi et al. and Kong et al. found that GRFs or a GRF–GIF chimeric protein dramatically improved the regeneration efficiencies in a broad range of dicot and monocot species [68,85]. More recently, Lian et al. showed that *PLETHORA5 (PLT5)* significantly improved shoot regeneration or somatic embryo formation and genetic transformation in snapdragon, tomato, cabbage, and pepper [86].

Epigenetic regulation is also critical for the acquisition of cell totipotency and the plant SE process, such as DNA methylation and histone modification, by affecting gene expression [19]. In cotton, the use of bisulfite-treated sequencing and a genome-wide single-base resolution methylation analysis has shown that non-embryogenic calluses (NECs) exhibit CHH hypomethylation compared with EC during the SE procedure, and the successive regenerated acclimation (SRA)-derived progeny of cotton Jin668 (regenerated plants, R4) are CHH hypomethylated compared with early regenerated plants (R0, R2) during the SRA process. Furthermore, the CHH methylation levels in the promoters of some hormone-related and WUSCHEL-related homeobox genes are negatively correlated with the expression levels of these genes during the SE process in cotton. Interestingly, DNA methylation inhibition in the cotton callus using zebularine treatment activates the expression of hormone-related tissue culture marker genes and increases the number of somatic embryos [87]. This suggests that DNA methylation may play a negative role in regulating SE.

Chromatin remodeling is a process that changes the chromatin structure between a condensed state and a transcriptionally accessible state, which is essential for controlling gene expression and SE [79,80,88]. From a number of studies, it has been concluded that histone acetylation is essential for the establishment of transcriptionally accessible chromatin [89]. Based on the transposase-accessible chromatin sequencing (ATAC-seq) method, scientists have described the chromatin accessibility landscape for plant SE and have shown that peak accessibilities are positively correlated with gene expression and the regeneration rate [79,80]. The histone deacetylase chemical inhibitor trichostatin A (TSA) reportedly exerts an improved SE effect by increasing the level of histone acetylation, inducing a

more open chromatin conformation, facilitating transcription complex interaction with some potential regeneration gene promoters, and activating gene expression [79,90,91]. Consistently, mutants of histone deacetylases have shown the formation of embryo-like structures on the true leaves of 6-week-old plants [92].

In conclusion, SE progress is a complicated regulatory network composed of hormone pathways, SE-related transcription factors, and epigenetic regulators. It has been proposed that optimizing the concentrations of various hormones in the medium, ectopic overexpression of SE-related transcription factors, and genetic disruption of epigenetic regulators would initiate somatic cell reprogramming to totipotent cells and induce SE [16].

## 7. Conclusions

Although SE technology can provide technical support for superior seedling propagation and genetic improvement of *H. brasiliensis*, challenges remain in the use of this technology. The SE ability is genotype-dependent, which is the major bottleneck limiting the wide application of SRJC micropropagation technology and genetic transformation technology in *H. brasiliensis*. To address this issue, we propose the following suggestions.

First, to properly address genotype-dependent issues, we need to acquire a better understanding of the mechanisms underlying the SE of *H. brasiliensis*. Recently, progress has been made in understanding the molecular mechanisms of SE in *H. brasilienesis* through differentially expressed gene analysis and homology-based cloning [93,94]. Several critical genes have been identified as differentially expressed in the embryogenic regenerating line or as highly expressed in the SE process, such as MADS-box genes and AP2/ERF genes, but the function of these genes has not been investigated [93,95,96]. Future research should attempt to explore the whole process and molecular regulatory mechanism of SE, including how immature seed inner integument or anther somatic cells can give rise to embryogenic cells through cell division and differentiation, what are the differences between the embryogenic cell and somatic cell at transcriptional, post-transcriptional, and epigenetic levels, what are the key genes that turn on cell totipotency and how can these genes regulate the cellular reprogramming of somatic cells toward the embryogenic pathway, how hormones cooperatively regulate cell totipotency and SE in *H. brasiliensis*, and what are the physiological and genetic differences in varieties with different SE capabilities. The global analysis of genomic, transcriptomic, epigenetic, proteomic, and metabolic profiling can be used to identify target genes that might regulate SE and to clarify the molecular mechanisms that promote SE in *H. brasiliensis*. In addition, the CRISPR/Cas9-based mutagenesis approach that has been set up for genome-wide targeted mutagenesis in rice and human cells can be developed in *H. brasiliensis* [97–99], and when coupled with high-throughput approaches, it will usher in a new era in which we can investigate the function of a series of SE-related genes.

Second, in order to promote the cell fate transition from somatic cells into somatic embryos, we should develop various optimized tissue culture systems for different varieties or generate developmental regulator overexpression lines. Recent studies have reported that regeneration and transformation efficiency have been improved by overexpressing developmental regulators in various genetic transformation-recalcitrant plant species and genotypes [68,81,85,100]. Therefore, the use of developmental regulator genes is a promising solution for improving SE. Once the molecular mechanism of SE in *H. brasiliensis* is better understood, genetic modification techniques can be used to regulate SE-related regulators' expression. Meanwhile, the combination of developmental regulators with various CRISPR/Cas system-based gene-targeting tools could be applied to more efficient genome editing [84,101].

Third, because successful applications of transgenic technology in plants depend on the efficient delivery of exogenous genes or gene-editing tools into cells, it is important to identify the most appropriate gene delivery method. Particle bombardment and the *Agrobacterium tumefaciens*-mediated transformation method are commonly used in genetic transformation, and the success of these methods strongly depends on the tedious

tissue culture system and is limited by genotypes. This indicates that the lack of appropriate delivery methods is also a barrier to the application of transgenic technology and Crispr/Cas-mediated gene editing in *H. brasiliensis*. Thus far, extensive attempts have been made to develop novel transformation protocols to improve gene delivery efficiency and simplify the tissue culture process. For example, to improve the infection efficiency of *Agrobacterium* and suppress plant defense response, researchers have developed an engineered *Agrobacterium tumefaciens* that expresses a type III secretion system (T3SS) from Gram-negative plant pathogenic bacteria *Pseudomonas syringae* to deliver *P. syringae* effectors that suppress plant defense responses and elevate gene delivery efficiency [102]. To do away with the tedious, time-consuming, and genotype-dependent tissue culture process, the floral dipping transformation protocol, the cut–dip–budding (CDB) delivery method, and the DNA-coated magnetic nanoparticle-mediated pollen transfection method combined with manual pollination have been developed [103–106]. Collectively, establishing a tissue-culture-free and genotype-independent transformation system in *H. brasiliensis* will facilitate a variety of improvements via genetic engineering and precision breeding.

**Author Contributions:** X.W. and X.G. designed the research and wrote the manuscript. Z.X. and Z.Y. collected images related to anther-derived somatic embryogenesis and genetic transformation. X.Y. and R.L. collected information related to genetic modifications in rubber trees and the field data. Q.Z. collected information related to cryopreservation in rubber trees. H.H. and T.H. conceived the manuscript. All authors have read and agreed to the published version of the manuscript.

**Funding:** This research was funded by Central Public-interest Scientific Institution Basal Research Fund for Chinese Academy of Tropical Agricultural Sciences (1630022023012); the Hainan Province Science and Technology Special Fund (ZDYF2021XDNY122); Special Fund for Hainan Excellent Team 'Rubber Genetics and Breeding' (20210203); and Hainan Provincial Natural Science Foundation of China (321MS0805).

**Data Availability Statement:** The data used to support the review are included within the article.

**Conflicts of Interest:** The authors declare no conflict of interest.

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
