# Peer review of "Current Achievements and Future Challenges of Genotype-Dependent Somatic Embryogenesis Techniques in Hevea brasiliensis"

_forests, doi:10.3390/f14091891_

Round 1

Reviewer 1 Report

1.Page No 15, Table 2, In column of somatic embryo induction rate  there will be ‘cotyledonary somatic embryos’ and not the ‘cotyledon somatic embryos’

line 356- F1 TO WRITTEN AS F1

Refrence no 4 delete the repeat word accessed

Author Response

We appreciate the valuable review comments! We have revised the manuscript following suggestion. Our point-by-point responses to the review comments are listed below.

Point 1: Page No 15, Table 2, In column of somatic embryo induction rate  there will be ‘cotyledonary somatic embryos’ and not the ‘cotyledon somatic embryos’.

Response 1: Thanks for pointing out the mistake. Following the suggestion, we have revised ‘cotyledon somatic embryos’ to be ‘cotyledonary somatic embryos’.

Point 2: 356 F1 TO WRITTEN AS F1. 

Response 2: Following the suggestion, we have revised ‘F1’ in line 356 to be ‘F1’.

Point 3: Refrence no 4 delete the repeat accessed.

Response 3: Thanks for pointing out the mistake, we have deleted the repeat “accessed”.

Reviewer 2 Report

1- The title is very long and needs editing

2- These authors contributed equally to this work. This part does not belong here...it is removed

3- Keywords are too long and inexpressive...need modification

4- It has been mentioned that there are many rubber-producing plants, please mention some of them as examples, with mentioning the differences in quality, quantities and areas of spread

5- Please always put the word in vitro in italics

6- Lardet et al. (2007), Zhou et al. (2012) We call attention to not using the method of listing references

7- The sentences must be linked as long as they are for the same researcher as in No. 12

8- (6±2°C) Such expressions need care when writing

9- As in Figure 1, each image represents a stage that takes a sub-number and is interpreted under the image

10- It is preferable to write mg / L in the form mg L -1

Minor editing of English language required

Reviewer 3 Report

The review provided important information on somatic embryogenesis and transformation to Hevea brasiliensis. English needs corrections. The main suggestions are found in the text. The article must be accepted after corrections.

The review provided important information on somatic embryogenesis and transformation to Hevea brasiliensis. English needs corrections. The main suggestions are found in the text. The article must be accepted after corrections.

Round 2

Reviewer 2 Report

 Accept in present form.